# FROM ENGLISH TO FOREIGN LANGUAGES: TRANSFERRING PRE-TRAINED LANGUAGE MODELS

## ABSTRACT

Pre-trained models have demonstrated their effectiveness in many downstream natural language processing (NLP) tasks. The availability of multilingual pre-trained models enables zero-shot transfer of NLP tasks from high resource languages to low resource ones. However, recent research in improving pre-trained models focuses heavily on English. While it is possible to train the latest neural architectures for other languages from scratch, it is undesirable due to the required amount of compute. In this work, we tackle the problem of transferring an existing pre-trained model from English to other languages under a limited computational budget. With a single GPU, our approach can obtain a foreign $BERT_{BASE}$ model within a day and a foreign $BERT_{LARGE}$ within two days. Furthermore, evaluating our models on six languages, we demonstrate that our models are better than multilingual BERT on two zero-shot tasks: natural language inference and dependency parsing.

## 1 INTRODUCTION

Pre-trained models (Devlin et al., 2019; Peters et al., 2018) have received much of attention recently thanks to their impressive results in many down stream NLP tasks. Additionally, multilingual pre-trained models enable many NLP applications for other languages via zero-short cross-lingual transfer. Zero-shot cross-lingual transfer has shown promising results for rapidly building applications for low resource languages. Wu & Dredze (2019) show the potential of multilingual-BERT (Devlin et al., 2019) in zero-shot transfer for a large number of languages from different language families on five NLP tasks, namely, natural language inference, document classification, named entity recognition, part-of-speech tagging, and dependency parsing.

Although multilingual models are an important ingredient for enhancing language technology in many languages, recent research on improving pre-trained models puts much emphasis on English (Radford et al., 2019; Dai et al., 2019; Yang et al., 2019). The current state of affairs makes it difficult to translate advancements in pre-training from English to non-English languages. To our best knowledge, there are only three available multilingual pre-trained models to date: (1) the multilingual-BERT (mBERT)[1] that supports 104 languages, (2) cross-lingual language model (XLM; Lample & Conneau, 2019)[2] that supports 100 languages, and (3) Language Agnostic SEntence Representations (LASER; Artetxe & Schwenk, 2019)[3] that supports 93 languages. Among the three models, LASER is based on neural machine translation approach and strictly requires parallel data to train.

*Do multilingual models always need to be trained from scratch? Can we transfer linguistic knowledge learned by English pre-trained models to other languages?* In this work, we develop a technique to rapidly transfer an existing pre-trained model from English to other languages in an energy efficient way (Strubell et al., 2019). As the first step, we focus on building a *bilingual* language model (LM) of English and a target language. Starting from a pre-trained English LM, we learn the target language specific parameters (*i.e.*, word embeddings), while keeping the encoder layers of the pre-trained English LM fixed. We then fine-tune both English and target model to obtain the bilingual LM. We apply our approach to autoencoding language models with masked language

---

[1] https://github.com/google-research/bert/blob/master/multilingual.md
[2] https://github.com/facebookresearch/XLM
[3] https://github.com/facebookresearch/LASER

model objective and show the advantage of the proposed approach in zero-shot transfer. Our main contributions in this work are:

- We propose a fast adaptation method for obtaining a bilingual BERT$_{\text{BASE}}$ of English and a target language within a day using one Tesla V100 16GB GPU.

- We evaluate our bilingual LMs for six languages on two zero-shot cross-lingual transfer tasks, namely natural language inference (XNLI; Conneau et al., 2018) and universal dependency parsing. We show that our models offer competitive performance or even better that mBERT.

- We illustrate that our bilingual LMs can serve as an excellent feature extractor in supervised dependency parsing task.

## 2 BILINGUAL PRE-TRAINED LMS

We first provide some background of pre-trained language models. Let $\boldsymbol{E}_e$ be English word-embeddings and $\Psi(\boldsymbol{\theta})$ be the Transformer (Vaswani et al., 2017) encoder with parameters $\boldsymbol{\theta}$. Let $\boldsymbol{e}_{w_i}$ denote the embedding of word $w_i$ (*i.e.*, $\boldsymbol{e}_{w_i} = \boldsymbol{E}_e[w_1]$). We omit positional embeddings and bias for clarity. A pre-trained LM typically performs the following computations: (i) transform a sequence of input tokens to contextualized representations $[\boldsymbol{c}_{w_1}, \ldots, \boldsymbol{c}_{w_n}] = \Psi(\boldsymbol{e}_{w_1}, \ldots, \boldsymbol{e}_{w_n}; \boldsymbol{\theta})$, and (ii) predict an output word $y_i$ at $i^{\text{th}}$ position $p(y_i|\boldsymbol{c}_{w_i}) \propto \exp(\boldsymbol{c}_{w_i}^{\top} \boldsymbol{e}_{y_i})$.

Autoencoding LM (BERT; Devlin et al., 2019) corrupts some input tokens $w_i$ by replacing them with a special token [MASK]. It then predicts the original tokens $y_i = w_i$ from the corrupted tokens. Autoregressive LM (GPT-2; Radford et al., 2019) predicts the next token ($y_i = w_{i+1}$) given all the previous tokens. The recently proposed XLNet model (Yang et al., 2019) is an autoregressive LM that factorizes output with all possible permutations, which shows empirical performance improvement over GPT-2 due to the ability to capture bidirectional context. Here we assume that the encoder performs necessary masking with respect to each training objective.

Given an English pre-trained LM, we wish to learn a bilingual LM for English and a given target language $\ell$ under a limited computational resource budget. To quickly build a bilingual LM, we directly adapt the English pre-traind model to the target model. Our approach consists of three steps. First, we initialize target language word-embeddings $\boldsymbol{E}_\ell$ in the English embedding space such that embeddings of a target word and its English equivalents are close together (§2.1). Next, we create a target LM from the target embeddings and the English encoder $\Psi(\boldsymbol{\theta})$. We then fine-tune target embeddings while keeping $\Psi(\boldsymbol{\theta})$ fixed (§2.2). Finally, we construct a bilingual LM of $\boldsymbol{E}_e$, $\boldsymbol{E}_\ell$, and $\Psi(\boldsymbol{\theta})$ and fine-tune all the parameters (§2.3). Figure 1 illustrates the last two steps in our approach.

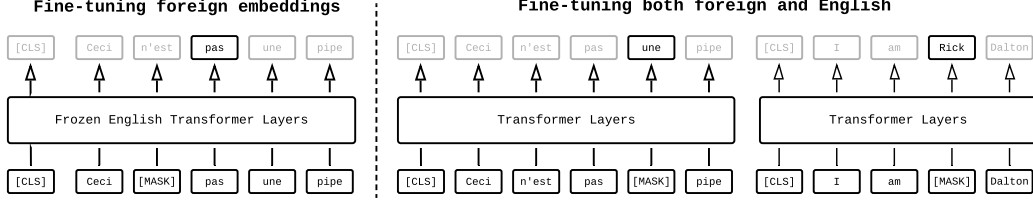

Figure 1: Pictorial illustration of our approach. Left: fine-tune language specific parameters while keeping transformer encoder fixed. Right: jointly train a bilingual LM and update all the parameters.

### 2.1 INITIALIZING TARGET EMBEDDINGS

Our approach to learn the initial foreign word embeddings $\boldsymbol{E}_\ell \in \mathbb{R}^{|V_\ell| \times d}$ is based on the idea of mapping the trained English word embeddings $\boldsymbol{E}_e \in \mathbb{R}^{|V_e| \times d}$ to $\boldsymbol{E}_\ell$ such that if a foreign word and an English word are similar in meaning then their embeddings are similar. Borrowing the idea of universal lexical sharing from Gu et al. (2018), we represent each foreign word embedding

$\boldsymbol{E}_\ell[i] \in \mathbb{R}^d$ as a linear combination of English word embeddings $\boldsymbol{E}_e[j] \in \mathbb{R}^d$

$$\boldsymbol{E}_\ell[i] = \sum_{j=1}^{|V_e|} \alpha_{ij} \boldsymbol{E}_e[j] = \boldsymbol{\alpha}_i \boldsymbol{E}_e \qquad (1)$$

where $\boldsymbol{\alpha}_i \in \mathbb{R}^{|V_e|}$ is a sparse vector and $\sum_j^{|V_e|} \alpha_{ij} = 1$.

In this step of initializing foreign embeddings, having a good estimation of $\boldsymbol{\alpha}$ could speed of the convergence when tuning the foreign model and enable zero-shot transfer (§5). In the following, we discuss how to estimate $\boldsymbol{\alpha}_i \; \forall i \in \{1, 2, \ldots, |V_\ell|\}$ under two scenarios: (i) we have parallel data of English-foreign, and (ii) we only rely on English and foreign monolingual data.

**Learning from Parallel Corpus**    Given an English-foreign parallel corpus, we can estimate word translation probability $p(e_j \,|\, \ell_i)$ for any (English-foreign) pair $(e_j, \ell_i)$ using popular word-alignment (Brown et al., 1993) toolkits such as fast-align (Dyer et al., 2013). We then assign:

$$\alpha_{ij} = p(e_j \,|\, \ell_i) \qquad (2)$$

Since $\boldsymbol{\alpha}_i$ is estimated from word alignment, it is a sparse vector.

**Learning from Monolingual Corpus**    For low resource languages, parallel data may not be available. In this case, we rely only on monolingual data (*e.g.*, Wikipedias). We estimate word translation probabilities from word embeddings of the two languages. Word vectors of these languages can be learned using fastText (Bojanowski et al., 2017) and then are aligned into a shared space with English (Lample et al., 2018b; Joulin et al., 2018). Unlike learning contextualized representations, learning word vectors is fast and computationally cheap. Given the aligned vectors $\bar{\boldsymbol{E}}_\ell$ of foreign and $\bar{\boldsymbol{E}}_e$ of English, we calculate the word translation matrix $\boldsymbol{A} \in \mathbb{R}^{|V_\ell| \times |V_e|}$ as

$$\boldsymbol{A} = \mathrm{sparsemax}(\bar{\boldsymbol{E}}_\ell \bar{\boldsymbol{E}}_e^\top) \qquad (3)$$

Here, we use $\mathrm{sparsemax}$ (Martins & Astudillo, 2016) instead of softmax. Sparsemax is a sparse version of softmax and it puts zero probabilities on most of the word in the English vocabulary except few English words that are similar to a given foreign word. This property is desirable in our approach since it leads to a better initialization of the foreign embeddings.

## 2.2 Fine-tuning Target Embeddings

After initializing foreign word-embeddings, we replace English word-embeddings in the English pre-trained LM with foreign word-embeddings to obtain the foreign LM. We then fine-tune *only* foreign word-embeddings on target monolingual data. The training objective is the same as the training objective of the English pre-trained LM (*i.e.*, masked LM for BERT). Since the trained encoder $\Psi(\boldsymbol{\theta})$ is good at capturing association, the purpose of this step is to further optimize target embeddings such that the target LM can utilized the trained encoder for association task. For example, if the words *Albert Camus* presented in a French input sequence, the self-attention in the encoder more likely attends to words *absurde* and *existentialisme* once their embeddings are tuned.

## 2.3 Fine-tuning Bilingual LM

We create a bilingual LM by plugging foreign language specific parameters to the pre-trained English LM (Figure 1). The new model has two separate embedding layers and output layers, one for English and one for foreign language. The encoder layer in between is shared. We then fine-tune this model using English and foreign monolingual data. Here, we keep tuning the model on English to ensure that it does not forget what it has learned in English and that we can use the resulting model for zero-shot transfer (§3). In this step, the encoder parameters are also updated so that in can learn syntactic aspects (*i.e.*, word order, morphological agreement) of the target languages.

## 3 Zero-shot Experiments

In the scope of this work, we focus on transferring autoencoding LMs trained with masked language model objective. We choose BERT and RoBERTa (Liu et al., 2019) as the source models

for building our bilingual LMs, named RAMEN[4] for the ease of discussion. For each pre-trained model, we experiment with 12 layers ($BERT_{BASE}$ and $RoBERTa_{BASE}$) and 24 layers ($BERT_{LARGE}$ and $RoBERTa_{LARGE}$) variants. Using $BERT_{BASE}$ allows us to compare the results with mBERT model. Using $BERT_{LARGE}$ and RoBERTa allows us to investigate whether the performance of the target LM correlates with the performance of the source pre-trained model. RoBERTa is a recently published model that is similar to BERT architecturally but with an improved training procedure. By training for longer time, with bigger batches, on more data, and on longer sequences, RoBERTa matched or exceed previously published models including XLNet. We include RoBERTa in our experiments to validate the motivation of our work: *with similar architecture, does a stronger pre-trained English model results in a stronger bilingual LM?* We evaluate our models on two cross-lingual zero-shot tasks: (1) Cross-lingual Natural Language Inference (XNLI) and (2) dependency parsing.

## 3.1 DATA

We evaluate our approach for six target languages: French (`fr`), Russian (`ru`), Arabic (`ar`), Chinese (`zh`), Hindi (`hi`), and Vietnamese (`vi`). These languages belong to four different language families. French, Russian, and Hindi are Indo-European languages, similar to English. Arabic, Chinese, and Vietnamese belong to Afro-Asiatic, Sino-Tibetan, and Austro-Asiatic family respectively. The choice of the six languages also reflects different training conditions depending on the amount of monolingual data. French and Russian, and Arabic can be regarded as high resource languages whereas Hindi has far less data and can be considered as low resource.

For experiments that use parallel data to initialize foreign specific parameters, we use the same datasets in the work of Lample & Conneau (2019). Specifically, we use United Nations Parallel Corpus (Ziemski et al., 2016) for `en-ru`, `en-ar`, `en-zh`, and `en-fr`. We collect `en-hi` parallel data from IIT Bombay corpus (Kunchukuttan et al., 2018) and `en-vi` data from OpenSubtitles 2018[5]. For experiments that use only monolingual data to initialize foreign parameters, instead of training word-vectors from the scratch, we use the pre-trained word vectors[6] from fastText (Bojanowski et al., 2017) to estimate word translation probabilities (Eq. 3). We align these vectors into a common space using orthogonal Procrustes (Artetxe et al., 2016; Lample et al., 2018b; Joulin et al., 2018). We only use *identical words* between the two languages as the supervised signal. We use WikiExtractor[7] to extract extract raw sentences from Wikipedias as monolingual data for fine-tuning target embeddings and bilingual LMs (§2.3). We *do not* lowercase or remove accents in our data preprocessing pipeline.

We tokenize English using the provided tokenizer from pre-trained models[8]. For target languages, we use fastBPE[9] to learn 30,000 BPE codes and 50,000 codes when transferring from BERT and RoBERTa respectively. We truncate the BPE vocabulary of foreign languages to match the size of the English vocabulary in the source models. Precisely, the size of foreign vocabulary is set to 32,000 when transferring from BERT and 50,000 when transferring from RoBERTa.

We use XNLI dataset (Conneau et al., 2018) for classification task and Universal Dependencies v2.4 (UD; Nivre et al., 2019) for parsing task. Since a language might have more than one treebank in Universal Dependencies, we use the following treebanks: `en_ewt` (English), `fr_gsd` (French), `ru_syntagrus` (Russian) `ar_padt` (Arabic), `vi_vtb` (Vietnamese), `hi_hdtb` (Hindi), and `zh_gsd` (Chinese).

**Remark on BPE** Lample et al. (2018a) show that sharing subwords between languages improves alignments between embedding spaces. Wu & Dredze (2019) observe a strong correlation between the percentage of overlapping subwords and mBERT's performances for cross-lingual zero-shot transfer. However, in our current approach, subwords between source and target are *not* shared. A subword that is in both English and foreign vocabulary has two different embeddings.

---

[4]The first author likes ramen.

[5]`http://opus.nlpl.eu/`

[6]`https://fasttext.cc/docs/en/crawl-vectors.html`

[7]`https://github.com/attardi/wikiextractor`

[8]`https://github.com/huggingface/pytorch-transformers`

[9]`https://github.com/glample/fastBPE`

## 3.2 ESTIMATING TRANSLATION PROBABILITIES

Since pre-trained models operate on subword level, we need to estimate subword translation probabilities. Therefore, we subsample 2M sentence pairs from each parallel corpus and tokenize the data into subwords before running fast-align (Dyer et al., 2013).

Estimating subword translation probabilities from aligned word vectors requires an additional processing step since the provided vectors from fastText are not at subword level[10]. We use the following approximation to obtain subword vectors: the vector $e_s$ of subword $s$ is the weighted average of all the aligned word vectors $e_{w_i}$ that have $s$ as an subword

$$e_s = \sum_{w_j : s \in w_j} \frac{p(w_j)}{n_s} e_{w_j} \tag{4}$$

where $p(w_j)$ is the unigram probability of word $w_j$ and $n_s = \sum_{w_j : s \in w_j} p(w_j)$. We take the top 50,000 words in each aligned word-vectors to compute subword vectors.

In both cases, not all the words in the foreign vocabulary can be initialized from the English word-embeddings. Those words are initialized randomly from a Gaussian $\mathcal{N}(0, 1/d^2)$.

## 3.3 HYPER-PARAMETERS

In all the experiments, we tune RAMEN$_{\text{BASE}}$ for 175,000 updates and RAMEN$_{\text{LARGE}}$ for 275,000 updates where the first 25,000 updates are for language specific parameters. The sequence length is set to 256. The mini-batch size are 64 and 24 when tuning language specific parameters using RAMEN$_{\text{BASE}}$ and RAMEN$_{\text{LARGE}}$ respectively. For tuning bilingual LMs, we use a mini-batch size of 64 for RAMEN$_{\text{BASE}}$ and 24 for RAMEN$_{\text{LARGE}}$ where half of the batch are English sequences and the other half are foreign sequences. This strategy of balancing mini-batch has been used in multilingual neural machine translation (Firat et al., 2016; Lee et al., 2017).

We optimize RAMEN$_{\text{BASE}}$ using Lookahead optimizer (Zhang et al., 2019) wrapped around Adam with the learning rate of $10^{-4}$, the number of fast weight updates $k = 5$, and interpolation parameter $\alpha = 0.5$. We choose Lookahead optimizer because it has been shown to be robust to the initial parameters of the based optimizer (Adam). For Adam optimizer, we linearly increase the learning rate from $10^{-7}$ to $10^{-4}$ in the first 4000 updates and then follow an inverse square root decay. All RAMEN$_{\text{LARGE}}$ models are optimized with Adam due to memory limit[11].

When fine-tuning RAMEN on XNLI and UD, we use a mini-batch size of 32, Adam's learning rate of $10^{-5}$. The number of epochs are set to 4 and 50 for XNLI and UD tasks respectively.

All experiments are carried out on a single Tesla V100 16GB GPU. Each RAMEN$_{\text{BASE}}$ model is trained within a day and each RAMEN$_{\text{LARGE}}$ is trained within two days[12].

## 4 RESULTS

In this section, we present the results of out models for two zero-shot cross lingual transfer tasks: XNLI and dependency parsing.

## 4.1 CROSS-LINGUAL NATURAL LANGUAGE INFERENCE

Table 1 shows the XNLI test accuracy. For reference, we also include the scores from the previous work, notably the state-of-the-art system XLM (Lample & Conneau, 2019). Before discussing the results, we spell out that the fairest comparison in this experiment is the comparison between mBERT and RAMEN$_{\text{BASE}}$+BERT trained with monolingual only.

---

[10]In our preliminary experiments, we learned the aligned subword vectors but it results in poor performances.

[11]Because Lookahead optimizer needs an extra copy of the model's parameters

[12]19 and 36 GPU hours, to be precise. Learning alignment with fast-align takes less than 2 hours and we do not account for training time of fastText vectors.

| | 🅰🈁 | fr | ru | ar | hi | vi | zh |
|---|---|---|---|---|---|---|---|
| Conneau et al. (2018) | ☐ | 67.7 | 65.4 | 64.8 | 64.1 | 66.4 | 65.8 |
| Artetxe & Schwenk (2019) | ☑ | 71.9 | 71.5 | 71.4 | 65.5 | 72.0 | 71.4 |
| Lample & Conneau (2019) (MLM) | ☐ | 76.5 | 73.1 | 68.5 | 65.7 | 72.1 | 71.9 |
| Lample & Conneau (2019) (MLM+TLM) | ☑ | 78.7 | 75.3 | 73.1 | **69.6** | 76.1 | 76.5 |
| mBERT (Wu & Dredze, 2019) | ☐ | 73.8 | 69.0 | 64.9 | 60.0 | 69.5 | 69.3 |
| RAMEN$_{\text{BASE}}$ | | | | | | | |
| + BERT | ☐ | 75.2 | 69.4 | 68.2 | 62.2 | 71.0 | 71.7 |
| | ☑ | 77.0 | 68.8 | 68.7 | 62.8 | 74.0 | 70.4 |
| + RoBERTa | ☐ | 79.2 | 72.4 | 71.6 | 63.4 | 74.9 | 73.3 |
| | ☑ | 78.2 | 73.1 | 72.0 | 65.0 | 73.7 | 74.2 |
| RAMEN$_{\text{LARGE}}$ | | | | | | | |
| + BERT | ☐ | 78.1 | 71.2 | 72.4 | 65.2 | 76.0 | 73.3 |
| | ☑ | 78.3 | 71.7 | 71.0 | 66.1 | 75.5 | 73.1 |
| + RoBERTa | ☐ | 81.7 | 73.6 | 72.8 | 64.0 | 79.2 | 74.1 |
| | ☑ | **81.7** | **76.4** | **75.0** | 68.5 | **79.7** | **77.7** |

Table 1: Zero-shot classification results on XNLI. ☑ indicates parallel data is used. RAMEN only uses parallel data for initialization. The best results are marked in **bold**.

We first discuss the transfer results from BERT. Initialized from fastText vectors, RAMEN$_{\text{BASE}}$ slightly outperforms mBERT by 1.9 points on average and widen the gap of 3.3 points on Arabic. RAMEN$_{\text{BASE}}$ gains extra 0.8 points on average when initialized from parallel data. With triple number of parameters, RAMEN$_{\text{LARGE}}$ offers an additional boost in term of accuracy and initialization with parallel data consistently improves the performance. It has been shown that BERT$_{\text{LARGE}}$ significantly outperforms BERT$_{\text{BASE}}$ on 11 English NLP tasks (Devlin et al., 2019), the strength of BERT$_{\text{LARGE}}$ also shows up when adapted to foreign languages.

Transferring from RoBERTa leads to better zero-shot accuracies. With the same initializing condition, RAMEN$_{\text{BASE}}$+RoBERTa outperforms RAMEN$_{\text{BASE}}$+BERT on average by 2.9 and 2.3 points when initializing from monolingual and parallel data respectively. This result show that with similar number of parameters, our approach benefits from a better English pre-trained model. When transferring from RoBERTa$_{\text{LARGE}}$, we obtain state-of-the-art results for five languages.

Currently, RAMEN only uses parallel data to initialize foreign embeddings. RAMEN can also exploit parallel data through translation objective proposed in XLM. We believe that by utilizing parallel data during the fine-tuning of RAMEN would bring additional benefits for zero-shot tasks. We leave this exploration to future work. In summary, starting from BERT$_{\text{BASE}}$, our approach obtains competitive bilingual LMs with mBERT for zero-shot XNLI. Our approach shows the accuracy gains when adapting from a better pre-trained model.

## 4.2 UNIVERSAL DEPENDENCY PARSING

We build on top of RAMEN a graph-based dependency parser (Dozat & Manning, 2016). For the purpose of evaluating the contextual representations learned by our model, we *do not* use part-of-speech tags. Contextualized representations are directly fed into Deep-Biaffine layers to predict arc and label scores. Table 2 presents the Labeled Attachment Scores (LAS) for zero-shot dependency parsing. Unlabeled Attachment Scores are provided in Appendix A.1.

We first look at the fairest comparison between mBERT and monolingually initialized RAMEN$_{\text{BASE}}$+BERT. The latter outperforms the former on five languages except Arabic. We observe the largest gain of +5.2 LAS for French. Chinese enjoys +3.1 LAS from our approach. With similar architecture (12 or 24 layers) and initialization (using monolingual or parallel data), RAMEN+RoBERTa performs better than RAMEN+BERT for most of the languages. Arabic and Hindi benefit the most from bigger models. For the other four languages, RAMEN$_{\text{LARGE}}$ renders a modest improvement over RAMEN$_{\text{BASE}}$.

| | 🅰🗷 | fr | ru | ar | hi | vi | zh |
|---|---|---|---|---|---|---|---|
| mBERT | ☐ | 71.6 | 65.2 | 36.4 | 30.4 | 35.7 | 26.6 |
| RAMEN$_{BASE}$ | | | | | | | |
|   + BERT | ☐ | 76.8 | 66.1 | 32.9 | 33.0 | 36.8 | 29.7 |
| | ☑ | 77.2 | 66.7 | 35.1 | 35.1 | 37.3 | 30.7 |
|   + RoBERTa | ☐ | 78.4 | 66.5 | 37.4 | 33.9 | 39.1 | 30.0 |
| | ☑ | 78.1 | 67.1 | 35.2 | 34.1 | 38.8 | 30.7 |
| RAMEN$_{LARGE}$ | | | | | | | |
|   + BERT | ☐ | 78.2 | 61.0 | 38.8 | 36.2 | 37.2 | 31.3 |
| | ☑ | 78.1 | 65.7 | 36.4 | 37.4 | 38.2 | 31.3 |
|   + RoBERTa | ☐ | 79.4 | **68.5** | 39.3 | 32.5 | **39.7** | **31.6** |
| | ☑ | **79.8** | 66.6 | **42.4** | **39.5** | 39.6 | 29.3 |

Table 2: LAS scores for zero-shot dependency parsing. ☑ indicates parallel data is used for initialization. Punctuation are removed during the evaluation. The best results are marked in **bold**.

## 5 ANALYSIS

### 5.1 IMPACT OF INITIALIZATION

Initializing foreign embeddings is the backbone of our approach. A good initialization leads to better zero-shot transfer results and enables fast adaptation. To verify the importance of a good initialization, we train a RAMEN$_{BASE}$+RoBERTa with foreign word-embeddings that are initialized randomly from $\mathcal{N}(0, 1/d^2)$. For a fair comparison, we use the same hyper-parameters in §3.3. Table 3 shows the results of XNLI and UD parsing of random initialization. In comparison to the initialization using aligned fastText vectors, random initialization decreases the zero-shot performance of RAMEN$_{BASE}$ by 15.9% for XNLI and 27.8 points for UD parsing on average. We also see that zero-shot parsing of SOV languages (Arabic and Hindi) suffers random initialization.

| | 🌱 | fr | ru | ar | hi | vi | zh | Δ |
|---|---|---|---|---|---|---|---|---|
| XNLI | rnd | 65.3 | 41.9 | 56.9 | 43.2 | 65.4 | 66.5 | |
| | vec | **79.2** | **72.4** | **71.6** | **63.4** | **74.9** | **73.3** | 15.9 |
| UD | rnd | 27.8 | 17.5 | 9.3 | 5.3 | 33.9 | 24.2 | |
| | vec | **78.4** | **66.5** | **37.4** | **33.9** | **39.1** | **30.0** | 27.8 |

Table 3: Comparison between random initialization (rnd) of language specific parameters and initialization using aligned fastText vectors (vec).

We also compare RAMEN to a bilingual BERT trained from scratch at 1,000,000 updates. On average, RAMEN outperforms bilingual BERT by 10.5% accuracy on XNLI task and 13.9 LAS on dependency parsing. The detail scores are provided in Appendix A.2.

### 5.2 ARE CONTEXTUAL REPRESENTATIONS FROM RAMEN ALSO GOOD FOR SUPERVISED PARSING?

All the RAMEN models are built from English and tuned on English for zero-shot cross-lingual tasks. It is reasonable to expect RAMENs do well in those tasks as we have shown in our experiments. *But are they also a good feature extractor for supervised tasks?* We offer a partial answer to this question by evaluating our model for supervised dependency parsing on UD datasets.

| | | fr | ru | ar | hi | vi | zh |
|---|---|---|---|---|---|---|---|
| mBERT | | 92.1 | 93.1 | 83.6 | 91.3 | 62.2 | 85.1 |
| BERT | + RAMEN$_{BASE}$ | 92.2 | 93.1 | 83.8 | 92.1 | 63.4 | 84.4 |
| | + RAMEN$_{LARGE}$ | 92.2 | 93.6 | 84.6 | 92.3 | 64.4 | 85.3 |
| RoBERTa | + RAMEN$_{BASE}$ | 92.7 | 93.5 | 85.1 | 92.3 | 65.3 | 85.7 |
| | + RAMEN$_{LARGE}$ | 93.1 | 94.0 | 85.3 | 92.6 | 66.1 | 86.9 |

Table 4: Evaluation in supervised UD parsing. The scores are LAS.

We used train/dev/test splits provided in UD to train and evaluate our RAMEN-based parser. Table 4 summarizes the results (LAS) of our supervised parser. For a fair comparison, we choose mBERT as the baseline and all the RAMEN models are initialized from aligned fastText vectors. With the same architecture of 12 Transformer layers, RAMEN$_{\text{BASE}}$+BERT performs competitive to mBERT and outshines mBERT by +1.2 points for Vietnamese. The best LAS results are obtained by RAMEN$_{\text{LARGE}}$+RoBERTa with 24 Transformer layers. Overall, our results indicate the potential of using contextual representations from RAMEN for supervised tasks.

### 5.3 How does linguistic knowledge transfer happen through each training stages?

We evaluate the performance of RAMEN+RoBERTa$_{\text{BASE}}$ (initialized from monolingual data) at each training steps: initialization of word embeddings (0K update), fine-tuning target embeddings (25K), and fine-tuning the model on both English and target language (at each 25K updates). The results are presented in Figure 2.

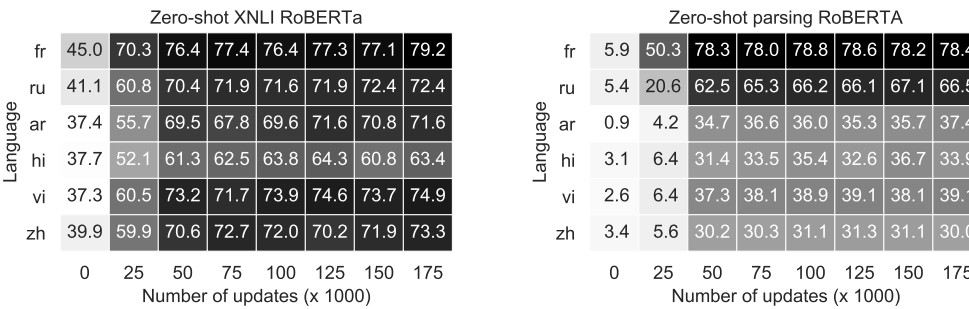

Figure 2: Accuracy and LAS evaluated at each checkpoints.

Without fine-tuning, the average accuracy of XLNI is 39.7% for a three-ways classification task, and the average LAS score is 3.6 for dependency parsing. We see the biggest leap in the performance after 50K updates. While semantic similarity task profits significantly at 25K updates of the target embeddings, syntactic task benefits with further fine-tuning the encoder. This is expected since the target languages might exhibit different syntactic structures than English and fine-tuning encoder helps to capture language specific structures. We observe a substantial gain in LAS from 25 to 40 LAS for all languages just after 25K updates of the encoder.

Language similarities have more impact on transferring syntax than semantics. Without tuning the English encoder, French enjoys 50.3 LAS for being closely related to English, whereas Arabic and Hindi, SOV languages, modestly reach 4.2 and 6.4 points using the SVO encoder. Although Chinese has SVO order, it is often seen as head-final while English is strong head-initial. Perhaps, this explains the poor performance for Chinese.

## 6 Conclusions

In this work, we have presented a simple and effective approach for rapidly building a bilingual LM under a limited computational budget. Using BERT as the starting point, we demonstrate that our approach performs better than mBERT on two cross-lingual zero-shot sentence classification and dependency parsing. We find that the performance of our bilingual LM, RAMEN, correlates with the performance of the original pre-trained English models. We also find that RAMEN is also a powerful feature extractor in supervised dependency parsing. Finally, we hope that our work sparks of interest in developing fast and effective methods for transferring pre-trained English models to other languages.

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

# A APPENDIX

## A.1 UNLABELED ATTACHMENT SCORES AND LABEL ATTACHMENT SCORES

| | A文 | fr | ru | ar | hi | vi | zh |
|---|---|---|---|---|---|---|---|
| mBERT | ☐ | 82.7/71.6 | 78.8/65.2 | 53.3/36.4 | 43.7/30.4 | 55.4/35.7 | 53.6/26.6 |
| RAMENBASE | | | | | | | |
| + BERT | ☐ | 84.1/76.8 | 80.2/66.1 | 50.4/32.9 | 46.0/33.0 | 56.1/36.8 | 59.4/29.7 |
| | ☑ | 84.5/77.2 | 79.5/66.7 | 51.9/35.1 | 49.4/35.1 | 56.9/37.3 | 60.1/30.7 |
| + RoBERTa | ☐ | 85.5/78.4 | 81.9/66.5 | 55.9/37.4 | 48.7/33.9 | 59.6/39.1 | 59.7/30.0 |
| | ☑ | 85.5/78.1 | 81.4/67.1 | 53.6/35.2 | 49.9/34.1 | 58.7/38.8 | 61.2/30.7 |
| RAMENLARGE | | | | | | | |
| + BERT | ☐ | 85.1/78.2 | 74.2/61.0 | 55.5/38.8 | 48.4/36.2 | 56.2/37.2 | 61.8/31.3 |
| | ☑ | 84.7/78.1 | 79.5/65.7 | 51.5/36.4 | 50.5/37.4 | 58.8/38.2 | 61.1/31.3 |
| + RoBERTa | ☐ | 86.5/79.4 | 82.8/68.5 | 53.6/39.3 | 49.3/32.5 | 58.9/39.7 | 61.0/31.6 |
| | ☑ | 86.5/79.8 | 81.5/66.6 | 58.6/42.4 | 56.0/39.5 | 59.7/39.6 | 60.2/29.3 |

Table 5: UAS/LAS scores for zero-shot dependency parsing. ☑ indicates parallel data is used for initialization. Punctuation are removed during the evaluation.

## A.2 COMPARISION TO BILINGUAL BERT

For each language pair, we train a bilingual BERT (b-BERT) from scratch using the hyperparameters provided in section §3.3. We learn 60,000 BPE codes on a concatenation of English and foreign language monolingual data. All b-BERT models are trained for 1,000,000 updates, which is significantly longer than RAMEN. Table 6 show the results of XNLI and UD.

| Task | Model | fr | ru | ar | hi | vi | zh | Δ |
|---|---|---|---|---|---|---|---|---|
| XNLI | b-BERT | 69.7 | 55.5 | 59.1 | 46.4 | 58.3 | 66.0 | |
| | RAMEN | **79.2** | **72.4** | **71.6** | **63.4** | **74.9** | **73.3** | **10.5** |
| UD | b-BERT | 69.2 | 41.0 | 17.8 | 14.9 | 26.3 | 22.6 | |
| | RAMEN | **78.4** | **66.5** | **37.4** | **33.9** | **39.1** | **30.0** | **13.9** |

Table 6: Comparison between b-BERT trained from scratch for 1,000,000 updates and RAMEN trained for 175,000 updates.

