# OpenReview forum: "From English to Foreign Languages: Transferring Pre-trained Language Models"
_ICLR.cc/2020/Conference — Reject_

### Official Review · AnonReviewer2 · 2019-10-23
**Official Blind Review #2**

**Rating:** 6

**Review:**

This paper proposes a method to adapt a pretrained BERT model from English to another languages with a limited time/GPU budget. Evaluation on 6 target languages shows good performance for natural language inference and dependency parsing.

Concretely, the proposed approach consists of, starting from a pretrained English language model, first training language-specific embeddings and then fine-tuning the entire pretrained model on English *and* the target language, using those embeddings. The language-specific embeddings are initialized based on the English embeddings (the authors propose two different ways for doing that).

I like about the paper that the approach is simple and fast. The experiments seem reasonable, too. The only minor negative point is that the approach is not particularly exciting.

**Experience Assessment:**

I have published one or two papers in this area.

**Review Assessment: Checking Correctness Of Derivations And Theory:**

I assessed the sensibility of the derivations and theory.

**Review Assessment: Checking Correctness Of Experiments:**

I assessed the sensibility of the experiments.

**Review Assessment: Thoroughness In Paper Reading:**

I read the paper at least twice and used my best judgement in assessing the paper.

---

> ### Author Response · Authors · 2019-11-12
> **Thank you for your feedback**
>
> We thank you for your feedback. We find the idea of transferring entire pretrained model is quite exiciting. When we started working on this idea, we try the most simple approach (presented in the paper) and to our surprise, it works quite well in comparision to multilingual BERT. We also observed that there is a large space for improvement in zero-shot transfer of dependency parsing. It seems that transferring syntax is harder than transferring semantics. We are looking forward to work on more exicting modeling approach to improve zero-shot dependency parsing.

---

### Official Review · AnonReviewer1 · 2019-10-23
**Official Blind Review #1**

**Rating:** 3

**Review:**

This paper presents a method to efficiently transfer pre-trained english language model to bilingual language model. The obtained representations are evaluated on downstream NLP task (natural language inference and dependency parsing) with state-of-the-art performances.


Pros:

- Experiments clearly show that, using the proposed method, stronger pre-trained English embedding leads to stronger bilingual language model and thus to better performances for downstream foreign tasks.

Cons:

While it is generally  intelligible, some structural modifications could be done to  improved the clarity of the paper. For instance, the method used to align foreign word vectors with English word vectors, when no aligned corpus is available, should appear sooner. It is described in 3.1 but should probably appear in 2.1 subsection Learning from Monolingual Corpus.

Minor issues:

- in section 3: RoBERA -> RoBERTa
- in section 5.1: the third sentence is syntactically incorrect
- in Conclusion: our approach produces better than -> our approach performs better than


**Experience Assessment:**

I have published one or two papers in this area.

**Review Assessment: Checking Correctness Of Derivations And Theory:**

N/A

**Review Assessment: Checking Correctness Of Experiments:**

I assessed the sensibility of the experiments.

**Review Assessment: Thoroughness In Paper Reading:**

N/A

---

### Official Review · AnonReviewer3 · 2019-10-24
**Official Blind Review #3**

**Rating:** 3

**Review:**

In this work, the authors propose a way to transfer a pre-trained English BERT model to a new language within a short amount of time. The key insight is to map English embeddings to the foreign language and have separate embeddings for both English and the foreign language. The resulting bilingual LM is evaluated for zero-shot transfer learning on two tasks: XNLI and dependency parsing.

Pros:
- The authors provide good details into their hyperparameter settings and about how the obtain the foreign language word embeddings.
- By leveraging existing pre-trained models, they’re able to do pre-training for their bilingual LM within 2 days.

Cons:
I find that a key comparison point in this paper is missing, which is Bilingual BERT trained on just the two languages that are being considered for their RAMEN system. This is not a fair comparison while mBERT which is trained on 100+ languages is not.
All comparisons are not fair since a simple baseline of just training mBERT on two languages with monolingual data and with a shared WPM is not evaluated here.
The proposed system has an unfair advantage over mBERT since it’s initialized from BERT/RoBERTA and fine-tuned only on two languages. Hence most of the parameters are used for just the two languages while mBERT uses the parameters for 104 languages.
Given this unfair comparison, I’m not sure if we can draw a meaningful conclusion from all the experiments.

Rating justification:
Given the lack a fair comparison between the bilingual and multilingual BERT models, I don't think the conclusions are insightful.


**Experience Assessment:**

I have published one or two papers in this area.

**Review Assessment: Checking Correctness Of Derivations And Theory:**

I carefully checked the derivations and theory.

**Review Assessment: Checking Correctness Of Experiments:**

I carefully checked the experiments.

**Review Assessment: Thoroughness In Paper Reading:**

I read the paper thoroughly.

---

> ### Author Response · Authors · 2019-11-12
> **comparison to Bilingual BERT**
>
> We thank reviewer for your valuable feedback. We agree with your point about the lack of comparision with a Bilingual BERT trained on two languages from scratch. We run an additional experiment to train a bilingual BERT-base from scratch for each language pair. For each language pair, we learn a join 60k bpe code and follow the same hyper-parameters setup described in section 3.3. In total, we train 6 bilingual BERT-base models. We evaluate these models at two checkpoints: (1) at 175,000 updates (the same number of updates with RAMEN), and (2) at 1,000,000 updates (~4 days) when we train these models longer.
>
> The results of XNLI (accuracy) and UD (Labeled Attachment Score) are listed bellow. We see that RAMEN outperform Bilingual BERT evaluated at the two checkpoints. Hoverever, we note that if the Bilingual BERT is trained on much longer time, perhap it will eventually match or surpass RAMEN performance.
>
> XNLI @ 175K updates
> fr: 60.8 | ru: 46.6 | ar: 48.7 | hi: 44.4 | vi: 52.1 | zh: 58.2
> XNLI  @ 1000K updates
> fr: 69.7 | ru: 55.5 | ar: 59.1 | hi: 46.4 | vi: 58.3 | zh: 66.0
> RAMEN - XNLI (copy from the paper for reference)
> fr: 75.2 | ru: 69.4 | ar: 68.2 | hi: 62.2| vi: 71.0 | zh: 71.7
>
>
> UD @ 175K updates
> fr: 61.1 | ru: 30.7 | ar: 14.7 | hi: 15.3 | vi: 24.5 | zh: 19.7
> UD  @ 1000K updates
> fr: 69.2 | ru: 41.0 | ar: 17.8 | hi: 14.9 | vi: 26.3 | zh: 22.6
> RAMEN - UD (copy from the paper for reference)
> fr: 76.8| ru: 66.1 | ar: 32.9 | hi: 33.0| vi: 36.8 | zh:  29.7

---

### Decision · Program_Chairs · 2019-12-19

**Decision:**

Reject

**Comment:**

This paper proposes a method to transfer a pretrained language model in one language (English) to a new language. The method first learns word embeddings for the new language while keeping the the body of the English model fixed, and further refines it in a fine-tuning procedure as a bilingual model. Experiments on XNLI and dependency parsing demonstrate the benefit of the proposed approach.

R3 pointed out that the paper is missing an important baseline, which is a bilingual BERT model. The authors acknowledged this in their rebuttal and ran a preliminary experiment to obtain a first set of results. However, since the main claim of the paper depends on this new experiment, which was not finished by the end of the rebuttal period, it is difficult to accept the paper in its current state. In an internal discussion, R1 also agreed that this baseline is critical to support the paper.

As a result, I recommend to reject this paper for ICLR. I encourage the authors to update their paper with the new experiment for submission to future conferences (given consistent results).

---

> ### Author Response · Authors · 2019-12-21
> **bilingual BERT baseline**
>
> I thank AC for their meta-review. However, I feel that the main point of the paper is missed by reviewers. The purpose of the proposed transfer approach is to obtain a good bilingual model under a limited computational budget (1 day for BERT based model). The extra experiments show that even with 4 days of training, the resulting bilingual BERT is still heavily underperformed the RAMEN model. If an unlimited computational resource is given, bilingual-BERT perhaps will match and outperform RAMEN. But this is not the point of the paper as stated in the abstract "While it is possible to train the latest neural architectures for other languages from scratch, it is undesirable due to the required amount of compute."